# Surface Functionalization of 4D Printed Substrates Using Polymeric and Metallic Wrinkles

**DOI:** 10.3390/polym15092117

**Published:** 2023-04-28

**Authors:** Johnson N. Agyapong, Bo Van Durme, Sandra Van Vlierberghe, James H. Henderson

**Affiliations:** 1The Bioinspired Institute, Syracuse University, Syracuse, NY 13244, USA; jagyapon@syr.edu (J.N.A.); bo.vandurme@ugent.be (B.V.D.); 2Department of Biomedical and Chemical Engineering, Syracuse University, Syracuse, NY 13244, USA; 3Polymer Chemistry and Biomaterials Group, Centre of Macromolecular Chemistry (CMaC), Department of Organic and Macromolecular Chemistry, Ghent University, 9000 Ghent, Belgium; sandra.vanvlierberghe@ugent.be

**Keywords:** 4D printing, wrinkles, surface functionalization, 3D printing, programming via printing, nuclear alignment

## Abstract

Wrinkle topographies have been studied as simple, versatile, and in some cases biomimetic surface functionalization strategies. To fabricate surface wrinkles, one material phenomenon employed is the mechanical-instability-driven wrinkling of thin films, which occurs when a deforming substrate produces sufficient compressive strain to buckle a surface thin film. Although thin-film wrinkling has been studied on shape-changing functional materials, including shape-memory polymers (SMPs), work to date has been primarily limited to simple geometries, such as flat, uniaxially-contracting substrates. Thus, there is a need for a strategy that would allow deformation of complex substrates or 3D parts to generate wrinkles on surfaces throughout that complex substrate or part. Here, 4D printing of SMPs is combined with polymeric and metallic thin films to develop and study an approach for fiber-level topographic functionalization suitable for use in printing of arbitrarily complex shape-changing substrates or parts. The effect of nozzle temperature, substrate architecture, and film thickness on wrinkles has been characterized, as well as wrinkle topography on nuclear alignment using scanning electron microscopy, atomic force microscopy, and fluorescent imaging. As nozzle temperature increased, wrinkle wavelength increased while strain trapping and nuclear alignment decreased. Moreover, with increasing film thickness, the wavelength increased as well.

## 1. Introduction

Wrinkle topographies have been studied as straightforward, elegant, easy to fabricate, versatile, and in some cases biomimetic surface functionalization strategies [1,2,3,4,5]. Application areas include metrology [6], optics [4], microfluidics [7], and cell mechanobiology [1,8]. In cell mechanobiology, wrinkle topographies on the micro- and nanoscale are used to study how cells respond to physical features of the extracellular environment and may mimic aspects of the extracellular matrix [1,5,8].

To fabricate surface wrinkles, one material phenomenon utilized is mechanical-instability-driven wrinkling. This phenomenon is observed in a bilayer system when a thin rigid film is compressed by contractile deformation of the underlying substrate [1,2,3]. The thin films exploiting this phenomenon have been metals [1,7,8,9], polymers [4,8], and nanomaterials [5] that have been introduced onto compliant substrates such as polydimethylsiloxane (PDMS) [2,9,10], hydrogels [11,12], and shape-memory polymers (SMPs) [1,13].

According to linear buckling theory, the minimum or critical strain ($\varepsilon_{c}$) required for buckling is governed by [1,2]
(1)$$\varepsilon_{c}=\frac{1}{4}{\left(\frac{{3\overset{-}{E}}_{s}}{{\overset{-}{E}}_{f}}\right)}^{\frac{2}{3}}$$
in which critical strain is dependent on the plane-strain modulus of the substrate (${\overset{-}{E}}_{s})$ and the plane-strain modulus of the thin film (${\overset{-}{E}}_{f}$), with the relationship between the plane-strain modulus ($\overset{-}{E}$), the Young’s modulus (E), and the Poisson’s ratio (v) being $\overset{-}{E}=E/(1-v^{2}$). For small deformations, the linear buckling theory predicts that the wrinkle wavelength (λ) is dependent on the film thickness ($h_{f})$, the Young’s modulus of the thin film (${\overset{-}{E}}_{f}$), and the modulus of the substrate (${\overset{-}{E}}_{s})$, whereas the wrinkle amplitude (A) is dependent on the film thickness ($h_{f})$, the applied strain (ε), and the critical strain ($\varepsilon_{c})$ [1,2,14,15] (Equations (2) and (3)).
(2)$$\lambda =2\pi h_{f}{\left(\frac{{\overset{-}{E}}_{f}}{{3\overset{-}{E}}_{s}}\right)}^{\frac{1}{3}}$$
(3)$$A=h_{f}{\left(\frac{\varepsilon }{\varepsilon_{c}}-1\right)}^{\frac{1}{2}}$$

To achieve the contractile substrate deformation necessary for wrinkle formation, strategies include thermal contraction [9,10], differential shrinkage [2,16], and mechanical actuation [1,2,4,6,17,18,19,20,21]. Compared to other strategies, mechanical actuation can offer more precise control over wrinkle formation via independent control of the timing, magnitude, and direction of the substrate deformation [3]. Yet, a practical challenge associated with the implementation of mechanical actuation is the need for an external apparatus, which precludes or complicates the use of the strategy in many environments, including cell culture and in vivo.

SMPs, a class of smart materials, have been used as the compliant substrate to eliminate the need for an external apparatus at the time of actuation [1,13]. SMPs can change shape on demand by ‘memorizing’ a permanent shape through crosslinking. They can be deformed and ‘programmed’ into a temporary shape by an immobilizing transition, such as crystallization or vitrification. Afterwards, they can recover into a permanent shape by a trigger, such as heat [1,22], light [23,24], electricity [25,26,27], magnetic field [28,29,30], water [31,32], or an enzyme [33]. For SMP-actuated wrinkling, the general strategy is to program an SMP in a temporary, elongated shape and coat the programmed SMP with a thin film so that, upon triggering, contraction of the SMP causes buckling of the film. This strategy has enabled the investigation of cell differentiation, morphology, motility, and tissue development in a biologically relevant mechanical environment [1,5,8,34].

Although the use of SMPs as the compliant substrate eliminates the need for an external apparatus *at the time of actuation*, the use of SMP-actuated wrinkling is itself limited by the need for a mechanical apparatus *at the time of shape-memory programming*. Even if dealing with a simple planar substrate, implementing an apparatus to create deformations more complex than uniaxial strain is beyond the capabilities of many labs. As a result, the study of SMP-actuated wrinkling has focused almost exclusively on uniaxial programming and strain recovery. Biaxial strains are possible but special rigs are required to program strains along multiple axes. If one considers the challenges associated with further translating the instability-driven wrinkling strategy to 3D substrates or scaffolds serving biomedical applications, it is clear that the apparatus required for programming would in many cases be laborious or impractical. There is, therefore, an unmet need for a strategy that would allow the deformation of complex substrates or 3D parts to generate wrinkles on surfaces throughout that complex substrate or part. If wrinkles could be integrated into complex platforms, they could enable the study of new wrinkling and mechanobiological phenomena and find a broader use in the control of cell behavior in tissue engineering and regenerative medicine strategies.

The goal of this work was to study a strategy suitable for functionalizing the surfaces of complex substrates or 3D parts with tunable polymeric and metallic wrinkles. To achieve this goal, we studied wrinkling on SMP 2D substrates prepared using a recently developed 3D printing strategy that traps strain in individual fibers and is, therefore, suitable for programming shape memory not only in 2D but also in 3D fabricated parts [35]. We studied the buckling of both gold (Au) and polystyrene (PS) thin films on the substrates. To demonstrate the functionality of the wrinkles in a model application, the effect of the wrinkle architecture and wrinkle topography on mammalian cell contact guidance was characterized using scanning electron microscopy (SEM), atomic force microscopy (AFM), and fluorescent imaging. To confirm translatability of the wrinkling approach to 3D fabricated parts, wrinkling was also demonstrated on the surface of cylinders composed of concentric fibers.

## 2. Materials and Methods

### 2.1. Experimental Design

To prepare SMP samples without the need for a separate programming step following fabrication, the approach was to use a 4D printing technique recently developed in our lab, known as programming via printing (PvP). Broadly speaking, 4D printing is 3D printing with functional materials to produce objects that change shape or function over time in response to an external stimulus, such as heat, light, water, or electricity [35,36,37,38,39,40,41,42]. The PvP technique specifically uses fused filament fabrication to fabricate an SMP part while simultaneously programming strains in the part at the fiber level [42] (Figure 1). Flat PvP substrates were coated with a thin rigid film (Au or PS) and subsequently recovered at 70 °C to induce wrinkle formation (Figure 1). The glass transition temperature (T_g_) and melting temperature (T_m_) of uncoated PvP substrates were characterized by differential scanning calorimetry (DSC). Uncoated samples were also studied to assess the effect of the nozzle temperature on strain trapping by calculating the percentage tensile strain (ε) along the length of the parallel fibers as
(4)$$\varepsilon =\frac{L_{o}-L_{f}}{L_{f}}\times 100$$
where *L_o_* = length before recovery and *L_f_* = length after recovery. Coated samples were used to study wrinkle formation through visualization and quantification in terms of wavelength and amplitude using SEM and AFM. Lastly, wrinkled samples seeded with cells underwent in vitro assays including live/dead and phalloidin and 4′,6-Diamidino-2-Phenylindole Dihydrochloride (DAPI) staining. To demonstrate the feasibility of wrinkle formation on 3D structures, wrinkles were induced along the lumen of a hollow tube.

### 2.2. Sample Preparation

Commercial thermoplastic polyurethane (TPU) SMP pellets (MM3520, SMP Technologies Inc., Tokyo, Japan) were dried for a minimum of 12 h in a vacuum oven at 50 °C before extrusion into filaments (Composer 450, 3Devo, Utrecht, The Netherlands). As a segmented polyurethane, this TPU is composed of a polyol soft segment and a diisocyanate and extender hard segment. The soft segment functions as the switching segment, which allows the programming of strain, whereas the hard segment acts as the net points, which facilitate ‘memory’ of the original shape [43]. To print the filament into desired shapes, STL files were converted to g-code (Cura, Ultimaker v. 4.13.1, Utrecht, The Netherlands) and printed using a 0.4 mm diameter nozzle (Ender 3 Pro, Creality, Shenzhen, China). Each fiber was printed at a height of 0.2 mm. To ensure surface smoothness and improve bed adhesion, the build plate was covered with Kapton tape and maintained at 25 °C. Square samples (10 mm × 10 mm, 1 mm thick) were printed for uniaxial wrinkle formation and solid cylinders (10 mm diameter, 2 mm thick) were printed for radial wrinkle formation. The square samples were printed as parallel fibers whereas the cylinders were printed as concentric fibers. The walls of the square substrate perpendicular to the fiber orientation were cut to prevent restriction of recovery. All samples were printed at 50 mm s^−1^ print speeds.

### 2.3. Thermal Characterization

The thermal properties of the TPU substrates were measured by DSC (Q200, TA Instruments, New Castle, DE, USA). An aluminum TZero pan was filled with 3 to 5 mg material and closed with an aluminum TZero lid. The sample pan and the reference pan were equilibrated at −40 °C and kept isothermal for 2 min. The temperature was increased to 210 °C with a constant rate of 10 °C min^−1^ and kept isothermal for 2 min. Next, the temperature was decreased to 0 °C at a constant rate of 10 °C min^−1^ and held isothermally for another 2 min. Finally, both pans were heated to 210 °C at a constant rate of 10 °C min^−1^. The T_g_ and T_m_ were determined using Universal Analysis Software (TA Instruments) as the change in slope during heating and the peak during an endothermal event, respectively.

### 2.4. Analysis of Strain Trapping in PvP Substrates

In PvP, the ability to draw (stretch) a fiber as it is deposited during fused filament fabrication is exploited to program tensile strains in the SMP print fiber-by-fiber, thereby combining fabrication and strain programming in a single step [42]. Fiber drawing is controlled via printing parameters, with nozzle temperature being a key parameter. To characterize the effect of the nozzle temperature on the percentage trapped strain within the PvP substrates used in this study, square substrates (N = 5 per nozzle temperature) were printed at six different nozzle temperatures (190–240 °C at 10 °C increments). To facilitate the deposition of the TPU during printing, the lower limit of the nozzle temperature must exceed the T_m_. The lower limit of the nozzle temperature range (190 °C) was chosen to avoid poor fiber fusion observed at lower nozzle temperatures during preliminary experiments. The upper limit of the nozzle temperature range was selected to avoid significant thermal decomposition which was reported at 260 °C [31]. To induce shape recovery, the TPU was heated above its transition temperature which corresponds to its T_g._ After recovery, the percentage tensile strain (ε) of the square substrates was calculated (Equation (4)) following measurement by calipers of length before and after recovery.

### 2.5. Wrinkle Formation and Characterization

To achieve mechanical-instability-driven wrinkling, a thin layer of Au or PS was coated onto the surfaces of the 4D printed substrates via sputter-coating or spin-coating, respectively. To study the effect of the film thickness on wrinkle characteristics, the Au samples were sputter-coated at 45 mA for 40, 60, 80, or 100 s (Vacuum Desk V Sputter-coater, Denton Vacuum, Moorestown, NJ, USA) and the PS group samples were coated with 2, 3, 4, or 5 weight percentage (wt%) of PS in toluene at 3000 rpm for 1 min using a spin-coater (WS-650-23B, Laurell Technologies, Lansdale, PA, USA). In preliminary experiments, brief exposure to toluene was observed to have no plasticization effect on the substrates. For the PS samples, PS pellets (Scientific Polymer Products Inc, Ontario, NY, USA) with a molar mass of 210,000 gmol^−1^ (GPC) were used as received. The coated substrates were recovered at 70 °C for 15 min in an isothermal oven to induce wrinkling. To visualize and characterize wrinkles, scanning electron microscopy (SEM, JSM-IT100LA, JEOL USA Inc. Peabody, PA, USA) and atomic force microscopy (AFM, Nano-R2, Pacific Nano Technology, Santa Clara, CA, USA) in tapping mode with a 5 N m^−1^ spring constant tip were utilized, respectively. Wavelength and amplitude were calculated from the surface profiles of AFM images using Gwyddion software (Gwyddion 2.60, Department of Nanometrology, Czech Metrology Institute). The wavelength was calculated as the peak-to-peak spacing whereas amplitude was calculated as the peak-to-trough distance (peak-to-peak amplitude).

### 2.6. In Vitro Cell Assays

To prepare samples for cell work, wrinkled samples were cut into 4 mm × 4 mm squares and sterilized for at least 2 h using a Thermo Scientific Biological Safety Cabinet’s built-in 365 nm UV bulb (Waltham, MA, USA). Next, the samples were soaked in cell culture media for a minimum of 2 h. Prior to cell experiments, C3H10T1/2 murine fibroblast cells (ATCC, Manassas, VA, USA) were expanded in Basal Medium Eagle (BME) supplemented with 10% fetal bovine serum, 2 mM l-glutamine, and 1% penicillin-streptomycin. At passages 10–15, cells were seeded at 800 cells mm^−2^ for experiments and incubated (37 °C; 5% CO_2_) for 24 h. Finally, a live/dead assay and 4′,6-Diamidino-2-Phenylindole Dihydrochloride (DAPI)/Phalloidin staining were performed to visualize double-stranded DNA and filamentous-actin (F-actin) in the cytoskeleton, respectively.

#### 2.6.1. Live/Dead Viability Assay

Calcein-acetoxymethyl (Calcein-AM, Invitrogen, Manassas, VA, USA), together with ethidium homodimer-1 (EthD-1, Invitrogen), were used to perform live/dead assays. Cells were exposed to a stock solution containing 1 μL Calcein-AM and 2 μL EthD-1 for every 2 mL of BME. After incubation (30 min; dark conditions), the solution was discarded, and the cells were visualized. An inverted fluorescence microscope (Leica DMI6000, Wetzlar, Germany) with LASX (Leica Application Suite X v. 3.7.5.24914) software was used for all imaging. Cell viability was quantified as the percentage of living cells relative to the total number of living and dead cells.

#### 2.6.2. DAPI and Phalloidin Assay

Cells were fixed using 4% formaldehyde, permeabilized using 0.1% Triton-X in phosphate-buffered saline (PBS), and blocked using bovine serum albumin, for 10, 5, and 30 min, respectively. Next, cells were exposed to a stock solution containing Alexa Fluor 546 conjugated phalloidin for 30 min. Afterwards, cells were washed with PBS followed by the addition of 300 nM DAPI stock solution for 5 min.

#### 2.6.3. Quantification of Nuclear Alignment

To quantify nuclear alignment, we used an approach we have previously described [1]. Briefly, ImageJ (v. 1.53t, Bethesda, MD, USA, http://imagej.nih.gov/ij (accessed on 23 April 2023)) was used to extract nuclear angles (ranging from 0° to 180° relative to an arbitrary reference angle of 0°) from DAPI-stained fluorescent micrographs as described [1]. Using an automated contour-based tracking algorithm [44], nuclear angles were normalized to the wrinkle angle and the truncated standard deviation (TSD, a measure of angular spread that quantifies alignment) was determined. The normalized nuclear angles were plotted as angular histograms [45]. A TSD of 52° denotes random alignment, whereas an angular spread of 0° would denote perfect alignment. To determine the significance of the nuclear alignment observed, statistical analysis was conducted on the angular spread of experimental and control groups.

### 2.7. Statistics

The Holm–Sidak multiple-comparison *t*-test was performed to test for effects of nozzle temperature and thin film thickness (Au sputter-time or PS weight percentage) on wrinkle wavelength and amplitude and for effects of nozzle temperature on cell viability and cell angular spread. Welch’s unequal variances *t*-test was performed to test for an effect of wrinkles on cell alignment, with all wrinkled groups pooled and compared to the flat tissue culture PS control group. Significance was set as *p* < 0.05. Origin (OriginLab v. 9.8.5.212, Northampton, MA, USA) software was used for all statistical tests. Data are reported as mean ± SD. Sample size is reported in the Section 3 of each experiment.

## 3. Results

### 3.1. Thermal Behavior of TPU and the Trapped Strain within PvP Substrates

DSC revealed the T_g_ of the TPU to be 35 °C, which is comparable to the value reported by the manufacturer, 35 °C (Figure 1). The T_g_ of the TPU was important in the present work, as it serves as the recovery temperature of the material. Although the manufacturer does not report the TPU’s T_m_, a T_m_ of 160 °C was observed (Figure 1). The effect of the nozzle temperature on the percentage trapped strain showed that an increase in nozzle temperature corresponded to a decrease in the percentage trapped strain (Figure 2). In addition to the contraction observed along the length of the fibers, expansion was observed in the width and height of substrates due to the Poisson effect.

### 3.2. Effect of Nozzle Temperature and Film Thickness on Wrinkle Wavelength and Amplitude

When the effect of nozzle temperature—a proxy for percentage trapped strain—on wrinkle properties was studied, it was found that as nozzle temperature increased (and trapped strain decreased) Au and PS wrinkle wavelength increased (Figure 3A,C). For example, 5 wt% PS wrinkle wavelength increased from 5.50 ± 1.65 μm at 200 °C to 9.71 ± 2.15 μm at 220 °C, and Au wrinkle wavelength increased from 1.91 ± 0.86 μm at 200 °C to 8.00 ± 2.66 μm at 240 °C for 100 s (Figure 3C). In contrast to the clear, statistically significant relationship between nozzle temperature (trapped strain) and wrinkle wavelength, few groups showed a statistically significant relationship between nozzle temperature (trapped strain) and amplitude, and no trends in the data were apparent (Figure 3B,D).

When the effect of film thickness was examined, it was found that as Au-sputter-time or PS wt% increased, wrinkle wavelength and amplitude also increased (Figure 3). Since Au (79 GPa) and PS (~1–3 GPa) have Young’s moduli that differ by orders of magnitude at 240 °C [4,46,47], a temperature at which the substrates possessed trapped strain of 3.83 ± 0.27%, no wrinkles were observed on the PS-coated samples, whereas wrinkles were still present on the Au-coated samples (Figure 3, Equation (1)).

### 3.3. Cytocompatibility and Nuclear Alignment of Wrinkled PvP Substrates

After 24 h, fluorescent micrographs of C3H10T1/2 cells seeded on Au-coated samples (sputtered for 60 s) showed high viability (>90%) irrespective of nozzle temperature (Figure 4B, *p* > 0.05). PS-coated sample were also observed to be cytocompatible in preliminary experiments. Cell nuclei were significantly more aligned on wrinkled substrates than on flat tissue culture PS controls (*p* = 0.00317; Figure 5; N = 9 for the pooled 200, 220, and 240 wrinkled samples and N = 3 for the PS controls). Specifically, compared to the control group, cells seeded on wrinkled substrates had angular histograms with a narrow nuclear angle distribution (Figure 5A, right). In addition, nuclear distribution showed a non-significant broadening trend as nozzle temperature increased (corresponding to a decrease in trapped strain; Figure 5A, right).

### 3.4. Effect of Substrate Architecture on Wrinkle Morphology

In addition to the flat, uniaxially wrinkled substrates used to study the relationship between the print parameters and wrinkle morphology, thin film parameters and wrinkle morphology, and cytocompatibility, we qualitatively explored the effect of print geometry on wrinkle morphology. Wrinkles were observed to align perpendicular to the fiber orientation. On the surface of the Au and PS-coated square substrates with parallel fibers, uniaxial wrinkles were formed (Figure 6A), which was consistent with the samples used in the quantitative characterization work. In contrast, on the surface of cylinders composed of concentric fibers, radial wrinkles were formed (Figure 6B). Moreover, by using the relationship between substrate architecture and wrinkle orientation, it was found that longitudinal PS wrinkles could be generated along the lumen of a hollow cylinder composed of rings stacked on top of each other (Figure 6C), thereby demonstrating the induction of wrinkling within a 3D structure and conceptually illustrating the potential of the combined wrinkling-PvP approach to create biomimetic structures for applications such as blood vessel engineering.

## 4. Discussion

The results demonstrate a successful strategy for functionalizing the surfaces of 4D printed substrates with tunable polymeric and metallic wrinkles. The nozzle temperature during printing, which controls the magnitude of trapped strain, was found to affect wrinkle wavelength but had no significant effect on wrinkle amplitude. Film thickness was also found to affect wrinkle wavelength and, to a lesser extent, amplitude.

The effect of the nozzle temperature on strain trapping can be attributed to the molecular orientation and the cooling history [38]. According to Van Manen et al., substrates printed at higher nozzle temperatures take longer to cool, giving them more time to organize into energetically favorable orientations, leading to lower strain [38,48]. Compared to the strains achievable using a mechanical apparatus, strain trapping achieved by PvP is currently limited to a small range of strains, but we anticipate that PvP strain trapping can be further improved through optimization of other parameters that affect fiber drawing during printing, such as the ratio of volumetric flow to nozzle translational speed. The present work focused on nozzle temperature because, unlike the nozzle temperature, in preliminary experiments the effect of printing speed (nozzle translational velocity) on trapped strain did show small differences but not a clear trend.

The wrinkling phenomenon on PvP substrates first demonstrated in this work can be tuned further in the future by additional modification of the film properties. In the present work, the observed effect of film type on wrinkle wavelength was expected, due to the difference in moduli of PS (~1–3 GPa) and Au (79 GPa) as well as the changes in film thickness associated with the Au sputter-time and PS weight percentage [4,46,47]. According to the linear buckling theory, wavelength is directly proportional to film thickness (Equation (2)) [1,2,14,15]. Although the film thicknesses were not directly measured in the present work, literature has shown that Au and PS thickness increase with increasing sputter-time and weight percentage, respectively [1,4]. As such, the difference in wrinkle wavelengths observed for different Au sputter-times or PS weight percentages at a given nozzle temperature can, therefore, be reasonably attributed to differences in film thickness.

It is worth noting that, although we expect fiber drawing and strain programming to be occurring at temperatures close to that of the nozzle (above both the material’s T_g_ and T_m_), the fiber can cool quickly during deposition. In prior reports, shape memory programming at relatively low temperatures, which in the present work would correspond to temperatures closer to the T_g_ than the T_m_, has been shown to affect strain programming [49]. Although such programming at lower temperatures generally has not been shown to affect the shape recovery, it has been found to significantly affect stress recovery. As such, in the future, examination and control of the precise temperature at which fiber drawing and strain programming occur may provide an additional level of control of trapped strain and stress energy and, consequently, the precise performance and functional capacity of PvP parts.

As has been observed with wrinkles induced after conventional strain programming, it was found that wrinkles on PvP substrates were cytocompatible and capable of altering nuclear orientation. Cell nuclei were significantly more aligned on wrinkled substrates than on flat tissue culture PS controls. In addition, nuclear distribution showed a non-significant broadening trend as nozzle temperature increased. Nuclear orientation serves as an indicator of cellular alignment due to the nucleo-cytoskeleton connection [50]. Regarding the observed non-significant trend, the ability of a surface pattern to direct cellular alignment is dependent on how the feature dimensions affect the attachment and spreading of subcellular focal adhesions [50,51]. Wrinkles with small wavelengths are densely packed, minimizing the contact area of the focal adhesions and forcing them to align to the wrinkles. Wrinkles with larger wavelengths, on the other hand, offer more space for the focal adhesions to spread, leading to the limited cellular alignment observed.

The effect of the wrinkles on cellular alignment is important, as cellular alignment has been shown to play a central role in embryogenesis [52] and tissue regeneration [53,54]. Cellular alignment also facilitates the cellular arrangement necessary for tissue function. Cell alignment is relevant to diverse cell processes; thus wrinkled PvP substrates can be anticipated to find use as an in vitro platform in the study of cell motility [44,55], morphology [1,12], differentiation [56], and topotaxis [57]. Additionally, the biocompatible T_g_ (trigger temperature) of the TPU used in this study would also facilitate the use of this approach in biomedical devices intended for actuation in vivo.

## 5. Conclusions

In this study, a strategy for functionalizing the surfaces of complex substrates or 3D parts with tunable polymeric and metallic wrinkles was demonstrated. The wrinkles can be tuned by controlling nozzle temperature, substrate architecture, film type, and film thickness. Lastly, the wrinkles are cytocompatible, inspiring potential applications as a novel straightforward, dynamic, and versatile in vitro cell platform.

## Data Availability

The data that supports the findings above can be requested from corresponding author.

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
