# Peer review of "Surface Functionalization of 4D Printed Substrates Using Polymeric and Metallic Wrinkles"

_polymers, 2023, doi:10.3390/polym15092117_

Round 1
Reviewer 1 Report
In this paper, the authors propose to deposit a gold or polystyrene layer on top of a 3D printed material that retracts after printing to generate wrinkles. Even though the concept is interesting and worth investigating, the work cannot be published in its current state.
1) The abstract is misleading. You say that there is a need for a strategy to generate wrinkles on complex substrates or 3D parts. Even though you study the process parameters that influence the wrinkles for a simple flat substrate, there is a huge lack of characterization and results when it comes to complex or 3D parts. Section 3.4 (see also point 6) is largely insufficient to justify your claim. You need to show a proper characterization of the wrinkles for an actual complex substrate or 3D parts. It is indeed unclear how you would control the wrinkles on such complex objects. Furthermore, the effect of wrinkles on cell alignment is disconnected from the topic announced in the title but takes a significant part of the paper. Hence, title, abstract and objectives need to be clarified, and possibly the manuscript reorganized, to be more consistent and tackle a clear scientific goal.
2) In the introduction, you need to introduce the seminal papers regarding wrinkles, buckling and their corresponding equations (e.g., articles by Bowden et al, 1998 and by those who pioneered the concept, i.e., JW Hutchinson and GM Whitesides). Your bibliography goes barely 10 years back.
3) In the introduction, there should be a wider bibliography on other ways to obtain wrinkles via non-contact actuations. For example, other smart materials such as magnetorheological elastomers have been employed for this purpose.
4) In the results, Section 3.2, check the precision of you instrument but it seems unlikely that you can get a 0.01°C precision on temperature.
5) In section 3.3, you do not say which parameters have been used for gold deposition. It does not make sense to present the results as a function of nozzle temperature, it needs to be translated in something that is characteristic of the wrinkles at hand. What is the size of the cells you deposit? Could you explain why you do not observe cell alignment for larger wavelength and compare to existing literature? Either you strongly make a case for the necessity of driving cell alignment and show convincing results, or this should not occupy too much of the article.
6) Section 3.4 is where we expect you to deliver on the promise of the abstract and introduction, i.e., uncover a strategy that functionalizes the surfaces of complex substrates or 3D parts with tunable polymeric and metallic wrinkles. You barely show an example on a cylinder where it is hard to understand from the caption (you need a scheme or arrows) what is the printing/fiber direction and the wrinkles direction relative to the part. There is no information on the wrinkles wavelength and amplitude even though they seem to play a key role in cell alignment. This section needs to be seriously expanded. Also, be careful with the use of the word “unable”, especially when 4D printing appears in the title: one expects wrinkles that can evolve after deposition.
Reviewer 2 Report
The presented work describing Surface Functionalization of 4D Printed Substrates Using Polymeric and Metallic Wrinkles is quite relevant, interesting and written in good scientific language. But I suggest Authors should revise the manuscript based on the points below before an acceptance.
The authors need to explain in more detail why the method they use is related to 4D printing, and not to 3D.
It is necessary to describe the fracture at in Figure 1 at a temperature of about 10 ℃
It would be useful to study and provide information on whether the C3H10T1/2 cells used are present on the surface of the sample (Figure 4) as a biofilm or as a cell suspension. For example, this could easily be done by laser confocal microscopy.
Information about the effect of surface morphology on the growth of microorganisms on it would also be important for the work. However, this aspect of the work can be left to the authors for future publications.
Reviewer 3 Report
The authors investigated using 3D-printed shape memory polymers to create wrinkled surfaces through shape recovery. The goal was to understand the relationship between printing temperature and wrinkle formation, including wrinkle wavelength and strain trapping. They also confirmed that the directionality of wrinkles impacts the growth behavior of cells and promotes their growth in alignment with the wrinkle direction. Overall, the manuscript was well written, gives enough experimental details, and adequately addresses the research question.
I have only a few minor suggestions to improve the quality of this manuscript:
1. Section 2.2 sample preparation could be complemented by images or figures to describe the printed patterns better. It is hard to visualize the patterns and shapes of the cylinders for radial wrinkle formation. Some pictures of the printed parts (or a schematic like for the parallel fibers in scheme 1) would be helpful for the reader.
2. I am curious if the authors have also looked at the printing speed as available in addition to different temperatures. You could add a paragraph about your thoughts on this in the discussion part.
3. Wrinkle formation and characterization: The authors report that the samples were coated with PS in toluene. Are there any concerns that the toluene causes plasticization of the SMP and therefore causes early recovery?
4. Discussion: Shape memory polymers can be functionalized through the Tg or/and Tm. Could the authors discuss which mechanism is utilized for their TPU during the printing process and explain the contribution of hard and soft segments toward this process? Not all readers might be familiar with this.
5. Discussion: The authors explain the different strain-trapping rates through cooling kinetics and time to relax polymer chains during the cooling process. I think it should also be considered what the nozzle temperature is with respect to the Tm of the polymer. Some of the samples could have been subjected to cold drawing (at least in portion) because the temperature of the polymers drops after leaving the nozzle and while being printed on the printing bed.
Round 2
Reviewer 1 Report
The authors significantly improved the manuscript both clarifying the objectives and providing new insightful discussions of their results, overall making the article much clearer.
I apologize for missing the seminal papers on wrinkling, but usually, seminal papers are expected to be cited as soon as the concept is introduced in order to highlight that this is the work from which the rest is derived.
I understand the point of the authors to present data as a function of nozzle temperature. However, it would be interesting in the future to provide information on film thickness as a function of deposition parameters. I would assume that in sputtering, identical deposition time may lead to different thicknesses depending on the equipment used.
Though obtaining wrinkles on 3D parts is no longer the central point of the article, it would have been interesting to provide a bit more information on the wavelength and amplitude of the wrinkles obtained in Fig. 6 but I assume this might be the focus of another work.